



# H$_2$O$_2$ modulates the energetic metabolism of the cloud microbiome

Nolwenn Wirgot[1], VirginieVinatier[1], Laurent Deguillaume[2], Martine Sancelme[1], and Anne-Marie Delort[1]

[1]Université Clermont Auvergne, CNRS, Sigma-Clermont, Institut de Chimie de Clermont-Ferrand, 63000 Clermont-Ferrand, France
[2]Université Clermont Auvergne, CNRS, Laboratoire de Météorologie Physique, 63000 Clermont-Ferrand, France

Correspondence to: A.-M. Delort (A-marie.delort@uca.fr)

**Abstract.** Chemical reactions in clouds lead to oxidation processes driven by radicals (mainly HO$^\bullet$, NO$_3$$^\bullet$ or HO$_2$$^\bullet$) or strong oxidants such as H$_2$O$_2$, O$_3$, nitrate and nitrite. Among those species, hydrogen peroxide plays a central role in the cloud chemistry by driving its oxidant capacity. In cloud droplets, H$_2$O$_2$ is transformed by microorganisms which are metabolically active. Biological activity can therefore impact the cloud oxidant capacity. The present article aims at highlighting the interactions between H$_2$O$_2$ and microorganisms within the cloud system.

First, experiments were performed with selected strains studied as reference isolated from clouds in microcosms designed to mimic the cloud chemical composition, including the presence of light and iron. Biotic and abiotic degradation rates of H$_2$O$_2$ were measured and results showed that biodegradation was the most efficient process together with photo-Fenton process. H$_2$O$_2$ strongly impacted the microbial energetic state as shown by adenosine triphosphate (ATP) measurements in the presence and absence of H$_2$O$_2$. This ATP depletion was not due to the loss of cell viability. Secondly, correlation studies were performed based on real cloud measurements from 37 clouds samples collected at the PUY station (1465 m a.s.l., France). The results support a strong correlation between ATP and H$_2$O$_2$ concentrations and confirm that H$_2$O$_2$ modulates the energetic metabolism of the cloud microbiome. The modulation of microbial metabolism by H$_2$O$_2$ concentration could thus impact cloud chemistry, in particular the biotransformation rates of carbon compounds and consequently can perturb the way the cloud system is modifying the global atmospheric chemistry.

Keywords: Cloud water, Microorganisms, Hydrogen peroxide, Energetic metabolism, Atmospheric chemistry

## 1 Introduction

The atmosphere is an oxidizing medium where trace gases are transformed/removed by oxidation including methane and other organic compounds, carbon monoxide, nitrogen oxides, and sulfur gases. Evaluating the oxidizing power of the atmosphere is crucial since it controls pollutant formation, aerosol production and greenhouse radiative forcing (Thompson, 1992).

In this context, hydroperoxides (ROOH) contribute to the oxidizing power of the troposphere (Lee et al., 2000) by controlling the cycling of HO$_x$ radicals (HO$^\bullet$, HO$_2$$^\bullet$). They can serve as temporary reservoirs of HO$_x$ radical since, for example, their photolysis and reactivity will regenerate HO$^\bullet$ radicals. Among hydroperoxide, hydrogen peroxide is a key gas phase atmospheric chemical species (Vione et al., 2003) with concentration in the gas





phase in the ppb$_v$ level or less. The atmospheric concentration of $H_2O_2$ is impacted by a variety of meteorological
parameters (*e.g.* actinic flux, temperature and relative humidity) and is affected by the levels of chemical species
such as VOCs, CO, $O_3$, and $NO_x$ (Lee et al., 2000). One of the significant parameter controlling the evolution of
$H_2O_2$ concentration is the actinic flux intensity. Diurnal and seasonal variations of hydrogen peroxide are shown
by field measurements with higher concentrations during the day and in summer than during the night and in
winter. This is linked to the atmospheric formation of $H_2O_2$ that results from a series of photochemical reactions
creating free radicals followed by corresponding radical reactions with appropriate precursor substances.
In the presence of atmospheric liquid water (cloud, fog, rain), $H_2O_2$ is rapidly dissolved because of its high
Henry's law constant (7.7 $10^4$ M/atm at 298K; Sander, 2014). In this liquid phase, it is also produced by aqueous
phase reactivity (Möller, 2009). Several field campaigns have reported $H_2O_2$ concentrations in atmospheric water
in the μM range (Gunz and Hoffmann, 1990; Marinoni et al., 2011; Deguillaume et al., 2014). Hydrogen
peroxide plays a central role in various important chemical processes in clouds. First, $H_2O_2$ is considered as the
most important oxidant for the conversion of sulfite to sulfate for pH lower than 5.5, therefore contributing
significantly to the acidification of clouds and precipitations (Deguillaume et al., 2004). Second, the photolysis
of $H_2O_2$ will lead to an efficient production of the hydroxyl radical $HO^\bullet$ and recent study have shown that this
can be a dominant aqueous source (Bianco et al., 2015). They can also directly oxidize organic compounds in the
aqueous phase (Schöne and Herrmann, 2015). Finally, $H_2O_2$ is involved in redox processes leading to the
conversion of reactive free radicals and trace metals such as iron (Kieber et al., 2001; Deguillaume et al., 2005).
Consequently, $H_2O_2$ is a key chemical compound controlling the aqueous phase oxidant capacity and leading to
the transformation of inorganic and organic compounds present in the atmospheric aqueous phase. The resulting
inorganic and organic products can contribute to the aerosol phase when the cloud evaporates leading to climatic
effect.
A few decades ago, microorganisms have been found alive in cloud water (Sattler et al., 2001; Amato et al.,
2005, 2007a,b). Particularly through measurements of adenosine triphosphate (ATP) and anabolic precursors or
nutrients incorporation rates, it has been shown that cloud microorganisms are metabolically active and play an
important role in cloud chemical reactivity (Sattler et al., 2001; Amato et al., 2007a; Hill et al., 2007;
Vaïtilingom et al., 2012, 2013). Few studies performed on simplified or real microcosms have demonstrated that
cloud microorganisms are able to degrade carbon compounds (Ariya et al., 2002; Amato et al., 2005, 2007c;
Husarova et al., 2011; Vaïtilingom et al., 2010, 2011, 2013; Matulovà et al., 2014).
Microorganisms are also in direct interaction with oxidant species in clouds (iron, hydroxyl radical, hydrogen
peroxide, *etc.*). Vaïtilingom et al. (2013) have demonstrated that microorganisms present in real cloud water are
able to degrade efficiently hydrogen peroxide. This suggests that cloud microorganisms have found strategies to
survive and resist to the stresses encountered in this medium and in particular to the oxidative stress. In this
context, Joly et al. (2015) have conducted laboratory experiments to investigate the survival of selected strains
(bacteria and yeasts) isolated from cloud waters, in the presence of various concentrations of hydrogen peroxide.
The results showed that the survival rates of the studied strains were not affected by $H_2O_2$ exposure. In addition,
the strains were exposed to artificial UV-visible light mimicking the natural solar irradiation inside clouds. No
significant impact on the survival of the bacterial strains was observed.
These results have been confirmed in real cloud water, including the microflora and chemical complexity (iron,
$H_2O_2$, *etc.*), incubated in a photo-bioreactor designed to mimic cloud conditions (Vaïtilingom et al., 2013).



Thanks to ADP/ATP ratio measurements, reflecting the energetic metabolism of microorganisms, exposed or not
to solar radiations, it has been shown that microorganisms were not impacted by artificial light and consequently
by the generation of radicals from $H_2O_2$ photo-reactivity. In addition, $H_2O_2$ is efficiently degraded by catalases
and peroxidases involved in the oxidative metabolism. Solar light did not modify the degradation rates of $H_2O_2$,
demonstrating that the biological process was not inhibited by UV radiations and radicals.
Indeed solar light can indirectly impact the viability of cells by the production of reactive oxygenated species
(ROS) including $HO^\bullet$ and $O_2^{\bullet-}$ radicals. The main sources of these radicals are $H_2O_2$ photolysis or Fenton and
photo-Fenton reactions involving iron (Fe) and $H_2O_2$. Most of these compounds can cross the cytoplasmic
membrane by diffusion. Aerobic microorganisms can also produce similar ROS during respiration. These
radicals can potentially damage the major cellular components such as proteins, DNA and lipids and lead to
cellular death. Thanks to the fact that microorganisms usually are protected against these ROS, they can
specifically modify their metabolism to face oxidative stress taking place in clouds. Therefore, microorganisms
utilize various mechanisms involved in the oxidative stress metabolism such as i) the production of pigments and
of antioxidant molecules (vitamins, glutathione, *etc.*) which can scavenge radicals or ii) the production of
specific enzymes such as superoxide dismutase which can transform $O_2^{\bullet-}$ into $H_2O_2$. $H_2O_2$ can be dismutated or
reduced respectively by catalases and other peroxidases (Delort et al., 2017).
The studies from Vaïtilingom et al. (2013) and Joly et al. (2015) highlighted the interactions between biological
activity and oxidants in clouds. In the present work, we artificially reproduced cloud conditions in microcosms to
study the biotic and abiotic transformation of $H_2O_2$ and, conversely, the impact of hydrogen peroxide on the
metabolism of cloud microorganisms. For this purpose, we decided to study individually the effect of parameters
interacting with $H_2O_2$: UV radiation, iron and bacteria. Under various experimental conditions, the degradation
rates of $H_2O_2$ were followed to highlight how individual parameters control its transformation. Moreover, the
impact of $H_2O_2$ on the energetic state of the bacterial cells was evaluated by measuring the ATP concentration
over time when the cells were exposed or not to $H_2O_2$. In order to confirm our laboratory results on the
interaction between microorganisms and $H_2O_2$, we performed a correlation analysis considering bio-physico-
chemical parameters measured in real cloud samples collected at the PUY station. This work will lead to a better
description of the mechanisms linking biological activity and cloud reactivity. This is crucial to consider all the
sinks and sources of $H_2O_2$, especially in atmospheric chemistry models, since $H_2O_2$ impacts a lot of atmospheric
relevant processes in the atmosphere.
**2 Material and methods**
**2.1 Bacterial strains and growth conditions**
Three bacterial strains belonging to the Gamma-Proteobacteria (*Pseudomonas graminis*, 13b-3, DQ512786;
*Pseudomonas syringae*, 13b-2, DQ512785) and Alpha-Proteobacteria classes (*Sphingomonas sp.*, 14b-5,
DQ512789) were selected as model strains for these experiments. Bacteria were grown in 10 mL of R2A
medium (Reasoner and Geldreich, 1985) under stirring (200 r.p.m) at 17°C for approximately 17 h, 24 h or 48 h,
depending on the strain. Cells in the exponential growth phase were collected by centrifugation for 3 min at
10481 g. The supernatant was removed and the bacterial pellet was suspended and washed twice with an
artificial cloud solution (2.2). The concentration of cells was estimated by optical density at 575 nm to obtain a



concentration close to $10^6$ cell mL$^{-1}$. Finally, the concentration of cells was precisely determined by flow
cytometry analysis (BD Facscalibur Becton-Dickinson; $\lambda_{exc}$ = 488 nm; $\lambda_{em}$ = 530 nm) using a method based on the
addition of a fluorochrome (SYBR-green) for their counting.

**2.2 Biodegradation assays**

Biodegradation assays were performed in marine artificial cloud solution that mimics real cloud conditions as
described in Vaïtilingom et al. (2011). Stock solutions were prepared with the following concentrations : 200 µM
for acetic acid (CH$_3$COOH; Acros organics), 145 µM for formic acid (HCOOH; Fluka), 30 µM for oxalic acid
(H$_2$C$_2$O$_4$;Fluka), 15 µM for succinic acid (H$_6$C$_4$O$_4$; Fluka), 800 µM for ammonium nitrate (H$_4$N$_2$O$_3$; Fluka), 100
µM for magnesium chloride hexahydrate (MgCl$_2$, 6H$_2$O; Sigma-Aldrich), 50 µM for potassium sulfate (K$_2$SO$_4$;
Fluka), 400 µM for calcium chloride dihydrate (CaCl$_2$, 2H$_2$O; Sigma-Aldrich), 2000 µM for sodium chloride
(NaCl; Sigma-Aldrich), 1100 µM for sodium hydroxide (NaOH; Merck), 315 µM for sulfuric acid (H$_2$SO$_4$;
Sigma-Aldrich). Finally, the obtained solution was adjusted to pH 6 and sterilized by filtration (Polyethersulfone
membrane, 0.20 µm; Fisher Scientific) before use. The artificial cloud water solution was ten times more
concentrated than a real cloud water solution in order to stabilize the pH. This was also the case for bacteria
concentration because the bacteria/substrate ratio should be kept identical to that of real cloud. Indeed, it has
been demonstrated that if this ratio is maintained, the degradation rate remains constant (Vaïtilingom et al.,

132     2010).

The equipment was sterilized by autoclaving at 121°C for 20 minutes and all manipulations were performed
under sterile conditions. Biodegradation assays consisted in marine artificial cloud solutions inoculated with
bacterial cells and incubated in a bioreactor (Infors HT Multitron II) at 17°C in the presence or absence of
hydrogen peroxide solution, of iron complex solution and under irradiation or obscurity condition. At regular
intervals, samples were taken and stored at -20 °C.
For irradiation condition the bioreactor was equipped with lamps that emit UV-radiation (Sylvania Reptistar;
15W; 6500K; UVA (up to 30%), UVB (up to 5%)) to mimic solar irradiation measured directly in clouds at the
PUY station (Fig. SM1). The incubation flasks were Pyrex crystallizers covered with a Pyrex filter and equipped
with Teflon tubes of 8 mm Ø plugged with sterile cotton, letting air and light pass. They contained 100 mL of
artificial cloud solution under agitation (130 r.p.m) and inoculated at $10^6$ bacterial cells per mL (Vaïtilingom et
al., 2013).
For dark condition the incubation flasks were amber Erlenmeyer flasks plugged with sterile cotton letting air
pass and containing 100 mL of artificial cloud solution also inoculated at $10^6$ cell mL$^{-1}$.
Hydrogen peroxide solution was prepared from a commercial solution (H$_2$O$_2$, 30%; not stabilized Fluka
Analytical). 1:1 stoichiometry iron complex solution was prepared from iron (III) chloride hexahydrate (FeCl$_3$,
6H$_2$O; Sigma-Aldrich) and from (S,S)- ethylenediamine-N,N'-disuccinic acid trisodium salt (EDDS, 35% in
water). The hydrogen peroxide solution and the iron complex solution were freshly prepared before each
experiment and the final working concentrations were fixed at 20 µM and 4 µM respectively, in agreement with
the real concentrations detected in samples collected at the PUY station (Deguillaume et al., 2014).



### 2.3 Analyses

Hydrogen peroxide was quantified with a miniaturised Lazrus fluorimetric assay (Lazrus et al., 1985; Vaïtilingom et al., 2013). This method is based on a reaction between hydrogen peroxide, Horse Radish Peroxidase (HRP) and 4-hydroxyphenylacetic acid that produces a fluorescent dimeric compound. Fluorescence readings (Safire II TECAN$^{©}$; $\lambda_{exc}$= 320 nm, $\lambda_{em}$= 390 nm) were made in a 96 well plate format.

Bioluminescence was used to analyse adenosine triphosphate (ATP) concentrations (Glomax® 20/20 single tube luminometer from Promega). This technique is based on an enzymatic reaction involving luciferin and luciferase. The protocol used was adapted from Biothema$^{©}$commercial kit instructions (Koutny et al., 2006).

In order to determine the survival rate of microorganisms in the presence of hydrogen peroxide (20 μM), plate-counts were performed on R2A agar medium at the beginning of each experiment and after 3, 7 and 24 hours of incubation. Plates were incubated 3 days at 17°C before CFU counts.

### 2.4 Determination of the initial degradation rates of hydrogen peroxide

The linear parts of kinetics were fit linearly (affine function) with the Origin 6.1 software. Three replicates were done. Error bars represent the standard errors (SEs) of the enzymatic assay (5%). In order to quantify the degradation rates of hydrogen peroxide in all studied conditions the initial degradation rate was used.

### 2.5 Statistical analysis

The R software was used to process the data in order to carry out principal component analysis (PCA). Statistical significance test was evaluated using PAST software (Hammer et al., 2001). Mean difference was considered to be statistically significant for a p-value inferior to 0.05.

### 2.6 Cloud sampling

Cloud water sampling was performed on the summit of the PUY station (summit of the puy de Dôme, 1465 m a.s.l., France) which is part of the atmospheric survey networks EMEP, GAW, and ACTRIS. The detachable part of the impactor was beforehand sterilized by autoclave at 121°C during 20 min and the fixed part was rinsed with alcohol at 70° just before sampling. The physical-chemical characteristics of the liquid cloud samples and the biological parameters were measured (concentrations of organic acids, inorganic ions, $H_2O_2$, Fe(II) and Fe(III), ATP, bacteria and fungi;  pH value… *etc.*). More information about the cloud sample collection is given in Deguillaume et al. (2014).

### 3 Results

The interactions between $H_2O_2$, which is one of the major oxidant present in clouds, and microorganisms were investigated by performing experiments in artificial cloud microcosms but also by considering chemical and biological parameters measured in real cloud samples over long period at the PUY station.

### 3.1 Experiments in artificial cloud water microcosms

Experiments were conducted in microcosms mimicking cloud conditions in which each important parameters including $H_2O_2$, iron, light and the presence of bacteria could be studied individually or in complementarity.



Microcosms consisted of photobioreactors containing artificial cloud water specially designed to be exposed to
artificial light which spectrum was as close as possible to the solar spectrum recorded under cloudy conditions
(Fig. SM1).
Artificial cloud water was mimicking cloud chemical composition from cloud samples classified as "marine"
following the work from Deguillaume et al. (2014) at the PUY station. The major part of the collected cloud
samples were classified as marine (52%) supporting our choice for the artificial cloud composition. The working
temperature was fixed at 17°C which is the average temperature of cloud samples in summer and the pH was
fixed at 6.0.
The three selected strains (*Pseudomonas* and *Sphingomonas*) were isolated from cloud water and are
representative of the genera most frequently found in cloud water samples (Vaïtilingom et al., 2012) collected at
the PUY site.
Hydrogen peroxide and iron complex (Fe-[EDDS]) were added or not to the solution in the incubators. These
two compounds are present in marine cloud water at average concentrations of 7.5 μM (with a dispersion of
mean values ranging from 0.1 – 20.8 μM) for $H_2O_2$ and 0.5 μM (with a dispersion of mean values ranging from
BDL. – 4.9) for Fe(III) (Deguillaume et al., 2014). In the cloud aqueous phase, Fe(III) may be complexed by
organic compounds. Recently, it has been hypothesized than iron can be chelated by other organic ligands of
biological origin (Herckes et al.., 2013; Herrmann et al., 2015), and in particular by siderophores (Vinatier et al.,
2016) that are ligands characterized by high complexing constants ($K>10^{20}$). Fe-[EDDS] was chosen as an
iron(III) complex model because this ligand has a complexing constant for iron very close to the values for
siderophores. Moreover, it is known to be stable at the working pH of 6.0 and because its chemistry has been
studied in details by Li et al. (2010).

**Hydrogen peroxide degradation in artificial cloud water**

$H_2O_2$ degradation was monitored periodically over a 8 h period. The kinetic profiles were similar for the three
strains. Results obtained with *Pseudomonas graminis* (13b-3) are illustrated in Figure 1 whereas the results
obtained with the other strains are presented for information in Figure SM2.
Under abiotic condition, the degradation of hydrogen peroxide is clearly effective in the presence of artificial
solar light and Fe-[EDDS] complex, due to the photo-Fenton reaction, with an initial degradation rate of 1.07 10$^{-9}$
mol L$^{-1}$ s$^{-1}$ (Table 1(a)). After 150 min this degradation rate decreases in parallel with EDDS by oxidation
occurs (Li et al., 2010). In the presence of the Fe-[EDDS] complex alone and in the absence of light, hydrogen
peroxide is almost not degraded. Indeed, the degradation rate of $H_2O_2$ due to the Fenton reaction is much lower
(2.23 10$^{-10}$ mol L$^{-1}$ s$^{-1}$) than the value obtained with the photo-Fenton reaction. Exposing the microcosm only to
our light conditions, the photolysis reaction of $H_2O_2$ is extremely slow (1.38 10$^{-10}$ mol L$^{-1}$ s$^{-1}$) due to the low
absorption of $H_2O_2$ in the solar spectrum measured inside a cloud and that was reproduced by the lamps used for
these experiments (Fig. SM1).
For the biotic conditions, three selected strains were tested: *Pseudomonas graminis* (13b-3), *Pseudomonas*
*syringae* (13b-2) and *Sphingomonas sp*. (14b-5). Initial biodegradation rates are summarized in Table 1(b).
These results show that, under our experimental conditions, hydrogen peroxide was degraded more efficiently in
the presence of bacteria even if the values obtained stay in the same order of magnitude compared to the abiotic
conditions with artificial light and Fe-[EDDS] complex. *Pseudomonas graminis* (13b-3) and *Pseudomonas*
*syringae* (13b-2) are the most active strains followed by *Sphingomonas sp* (14b-5). For each strain,



biodegradation rates are in the same order of magnitude without wide variations depending on the tested
conditions, *i.e.* in the presence or absence of artificial light and of Fe-[EDDS] complex.
These results show that artificial light and Fe-[EDDS] and thus $HO^\bullet$ radicals have no effect on $H_2O_2$
biodegradation. In addition, among the selected strains all degrade $H_2O_2$ in the same order of magnitude (average
value for the three strains and for the condition with iron and light $1.76\ 10^{-9}$ mol $L^{-1}$ $s^{-1}$ and with iron without
light $1.40\ 10^{-9}$ mol $L^{-1}$ $s^{-1}$). In Vaïtilingom et al. (2013), the same order of magnitude for the biodegradation rates
of $H_2O_2$ was found (average value for two distinct clouds with light $0.98\ 10^{-9}$ mol.$L^{-1}$ $s^{-1}$ and without light $0.29$
$10^{-9}$ mol $L^{-1}$ $s^{-1}$). The results obtained are in the same order of magnitude than values in real cloud environment
thereby validating our approach to analyse separately each parameter. This demonstrates that under our
experimental conditions, the selected strains degrade $H_2O_2$ like the microflora of real cloud.
**Impact of the $H_2O_2$ on the microbial energetic states in artificial marine cloud solution**
The impact of the presence of $H_2O_2$ on the energetic state of the bacterial cells was evaluated by measuring the
time evolution of ATP concentration for the three strains (Fig. 2). The ATP concentration was measured in the
presence (Fig. 2a, b, c - black square) or absence (Fig. 2a, b, c - white square) of $H_2O_2$. In the absence of $H_2O_2$, a
strong increase of ATP concentration was observed reflecting an active metabolism of the bacteria. On the
contrary, in the presence of $H_2O_2$, the results were clearly different and can be described in two phases. In the
first phase, ATP concentration was decreasing while in a second phase it was progressively increasing
(*Pseudomonas graminis,* 13b-3*)* or stabilizing (*Pseudomonas syringae,* 13b-2, *Sphingomonas sp,.* 14b-5). The
kinetics of ATP concentration evolution and $H_2O_2$ degradation are closely related. As discussed earlier (Fig. 1),
the $H_2O_2$ initially present (20 µM) was entirely degraded in approximately 3 h (depending on the strains); this
corresponds exactly to the end of the ATP decrease. Complementary experiments were performed with
incubations of the cells in the presence or absence of light and/or iron complex (Fe-[EDDS]) under conditions
similar to that described previously. The results obtained for the three strains are reported in Figure SM3
(*Pseudomonas graminis*), Figure SM4 (*Pseudomonas syringae*) and Figure SM5 (*Sphingomonas sp.*).
The results show that light and iron complex have no impact on the ATP concentration decrease. The measured
ATP concentration in the presence or absence of artificial light and/or iron(III) complex is similar to that
observed in the presence of $H_2O_2$ alone. The ATP concentration is thus only linked to the presence of $H_2O_2$.
**Impact of $H_2O_2$ on the survival of the microbial strains**
We also controlled that the decrease of ATP in the presence of $H_2O_2$ was not due to cell mortality. Samples of
artificial cloud medium inoculated at $10^6$ cell $mL^{-1}$ were incubated at 17°C with and without $H_2O_2$ (at 20 µM)
and the concentration of cells were determined by plate-counting. Figure 3 illustrates the results for all strains.
This figure shows the concentration of cells at different time of incubation for samples with or without $H_2O_2$.
The evolution of the cell concentration was not significantly different when cells were incubated in the presence
or absence of hydrogen peroxide. The decrease of ATP is therefore not linked to a lower concentration of cells
but to a modification of metabolic pathways due to $H_2O_2$ presence. The total number of cells was multiplied by a
factor 5 to 10 after 24h showing that bacteria were also able to divide and grow.





### 3.2 Impact of H₂O₂ on the microbial energetic metabolism in real cloud environment

In the previous section, we showed that $H_2O_2$ had a strong impact on the energetic metabolism of cells under our microcosm conditions. To go further, we looked at the potential impact of $H_2O_2$ on microbial energetic states in real cloud samples by carrying out statistical analyses based on data measured on real cloud water collected at the PUY station.

For this, principal component analysis (PCA) was used. In order to perform this multivariate statistical analysis, Table SM1 was built in such a way that lines and columns did not contain more than 10% of missing values. 37 clouds samples satisfied these criteria and were used for the PCA. These cloud events were collected between 2004 and 2013 at the PUY station. Various parameters were measured including ATP, bacteria and fungi concentration, inorganic and organic species concentration ($H_2O_2$, $SO_4^{2-}$, $NO_3^-$, $Cl^-$, acetate, formate, oxalate, $Na^+$, $NH_4^+$, $Mg^{2+}$, $K^+$, $Ca^{2+}$), temperature and pH (see Table SM1 for details). The origin of these clouds can be analyzed according to their back trajectories in four sectors (North/West, South/West, West and North/East). They can be also considered in four different categories considering their chemical composition (marine, continental, highly marine and polluted) as described in Deguillaume et al. (2014).

The result of the PCA analysis is presented in Figure 4. The first two dimensions contain practically 50% of the total inertia (total variance of the data table) reflecting the validity and reliability of the result. The PCA shows that if we consider all important parameters in the collected cloud samples a strong correlation appears between ATP and $H_2O_2$ concentrations (longer vectors and very close on the PCA). There is no correlation between ATP concentration and the number of bacteria (vectors practically orthogonal); this shows that $H_2O_2$ is linked to the energetic state of the cells and not to their concentration. Also, there is no correlation between ATP and markers of pollution such as the pH values, the $NO_3^-$, $SO_4^{2-}$ and $NH_4^+$ concentrations or even the temperature that could impact microbial metabolism.

In addition, Spearman rank correlation test (non-parametric test) was performed based on the 37 cloud samples to confirm the correlation between $H_2O_2$ and ATP. The values used for this test are presented in Table SM1. A p-value of 0.0047 was obtained with a Spearman's coefficient of 0.45 (Zar, 1972). This shows an extremely strong correlation between $H_2O_2$ and ATP as theoretically the Spearman's coefficient must be greater than 0.27 for 37 observations and the p-value inferior to 0.05 (significance threshold). To confirm that, ATP depletion due to $H_2O_2$ impact was not linked with the mortality of cells, a Spearman rank correlation test was also performed to evaluate the correlation between ATP and total microorganisms concentrations (sum of bacteria and fungi concentrations in Table SM1) (p-value superior to 0.37).

Figure 4 suggested that ATP or $H_2O_2$ could be also correlated to formate and oxalate as the vectors were relatively close. A Spearman rank correlation test (non-parametric test) was thus performed based on data extracted from the 37 cloud samples (Table SM1). A strong correlation was obtained between ATP and formate (p-value=0.0043, Spearman's coefficient = 0.46), and between $H_2O_2$ and formate (p-value = 0.00015, Spearman's coefficient= 0.58). ATP-oxalate correlation is rather weak (p-value = 0.030, Spearman's coefficient= 0.36) and much lower than the ATP-$H_2O_2$ correlation, similar values were obtained for oxalate and $H_2O_2$ (p-value = 0.035, Spearman's coefficient = 0.35).





**4 Discussion**
Our objective was to study in detail the interactions between cloud microorganisms and $H_2O_2$.
First we looked at the mechanisms involved in $H_2O_2$ transformations under laboratory conditions by isolating
each parameter to determine its impact on $H_2O_2$ (artificial light, Fe-[EDDS] complex and bacteria). Degradation
rates of hydrogen peroxide were precisely determined for different microbial strains frequently found in cloud
water samples collected to the PUY site. The results show that all bacterial strains studied under these conditions
degrade $H_2O_2$ in the same order of magnitude as abiotic conditions. The degradation rates of $H_2O_2$ by bacteria are
not impacted by the presence of light and Fe-[EDDS] and consequently by the generation of $HO^\bullet$ radicals. On
the opposite, in these laboratory experiments mimicking real cloud conditions, we have shown that $H_2O_2$ has a
strong impact on the microbial energetic state of the cells. This strong decrease of ATP concentration is not
linked to the number of cells as bacteria are able to divide and grow in the presence of $H_2O_2$. This reveals that
microorganisms are able to manage the stress induced by $H_2O_2$ through their metabolism in particular by the
involvement of enzymes (*e.g.* catalases, peroxidases, *etc.*) and other antioxidant molecules (glutathione, *etc.*). A
few studies report the decrease of ATP concentration in microorganisms (Perricone et al., 2003), plants (Tiwari
et al., 2002) or mammalian cells (Spragg et al., 1985; Josephson et al., 1991; Sporn and Peters-Goldenwhen,
1988, Hyslop et al., 1988; Oka et al., 2012) exposed to $H_2O_2$. Fig. 5 illustrates how $H_2O_2$ can affect the
concentration of ATP in the cells. First $H_2O_2$ can directly inhibit the ATP synthase, a membrane protein
synthetizing ATP from ADP (Tamarit et al 1998). Second $H_2O_2$ can impact different metabolic pathways which
are interconnected including glutathione metabolism, glycolysis, TCA cycle and DNA repair system. The
functioning of the enzymes in these pathways and also the activity of the ATP synthase are dependent on the
redox potential of the cells ($NAD^+$/NADH, $NADP^+$/NADPH ratios), and as a consequence the ATP
concentration is regulated by this redox potential (Haddock and Jones, 1977, Singh et al., 2007, Oka et al.,
2012). If for instance $NAD^+$ is depleted when the repair system is activated to avoid potential DNA damages
induced by $H_2O_2$, then ATP is depleted, and finally all the metabolic pathways involving these compounds are
impacted and a complete change in the metabolome can be expected.
We have shown, thanks to statistical analyses, that there was also a very strong correlation between $H_2O_2$ and
ATP concentrations in real cloud samples collected under various environmental conditions. We suggest thus
that hydrogen peroxide modulates the global metabolism of cloud microorganisms.
Another interesting correlation was obtained between $H_2O_2$ and formate as well as ATP with formate. This could
result from different concomitant processes. First formate is the most oxidized carbon molecule before $CO_2$
generated from successive oxidations of the organic matter by radicals issued from $H_2O_2$. Second it could reveal
the impact of $H_2O_2$ on the C1 metabolism; it is known that C1 compounds can be transformed by cloud
microorganisms (Husàrovà et al., 2011, Vaitilingom et al., 2010, 2011, 2013). In addition Thomas et al. (2016)
report the overproduction of formate in a strain of *Pseudomonas fluorescens* exposed to $H_2O_2$.
Finally, this work brings new insights into the interactions between $H_2O_2$ and the cloud microbiome and its
potential consequences on cloud chemistry (see Fig. 6).
First it confirms that cloud microorganisms are able to efficiently degrade hydrogen peroxide and potentially
impact the global carbon budget and the oxidant capacity of clouds as already shown in Vaïtilingom et al.
(2013). By decreasing $H_2O_2$ concentration, radical chemistry is less efficient to degrade the organic matter.
Second we show here for the first time that $H_2O_2$ impacts the energetic metabolism of the cloud microbiome and



thus potentially modulates its carbon metabolism. As a consequence it can modify the final transformation of the
organic matter in clouds. This reciprocal interaction between $H_2O_2$ and microorganisms and its subsequent
impact on cloud chemistry is clearly dependent on $H_2O_2$ concentration.
To go further in the understanding of the modulation of the metabolic pathways (including carbon, nitrogen,
amino-acids or sugars) induced by $H_2O_2$, a metabolomic approach could be used. The next step could be to
integrate biological data in numerical atmospheric models to better quantify consequence of this modulation on
atmospheric chemistry.
*Acknowledgments.* N. Wirgot is a recipient of a PhD fellowship from the MESR (French government).
Part of this work was supported by the French ANR program BIOCAP (ANR-13-BS06-0004). The authors are
very grateful to the OPGC/LaMP staff for collecting the cloud samples at the PUY station, see the database of
cloud biological and chemical composition wwwobs.univ-bpclermont.fr/SO/beam/index.php.

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



**Table 1: Initial rates of abiotic degradation (a) and of biotic degradation (b) of $H_2O_2$ measured in artificial cloud**
**water. Values are expressed in $10^{-9}$ mol $L^{-1}$ $s^{-1}$. Standard errors were calculated.**

| (a) | Light + Fe-EDDS] | Fe-[EDDS] | Light |
|---|---|---|---|
| | 1.07 | 0.22 | 0.14 |


| (b) | Light + Fe-[EDDS] + Bacteria | Fe-[EDDS] + Bacteria | Light + Bacteria | Bacteria |
|---|---|---|---|---|
| *Pseudomonas graminis* 13b-3 | 1.55 ± 0.25 | 1.93 ± 0.18 | 2.15 ± 0.018 | 2.07 ± 0.0093 |
| *Pseudomonas syringae* 13b-2 | 1.75 ± 0.15 | 1.27 ± 0.042 | 1.72 ± 0.14 | 1.18 ± 0.080 |
| *Sphingomonas sp.* 14b-5 | 1.97 ± 0.062 | 1.01 ± 0.21 | 0.87 ± 0.043 | 0.76 ± 0.11 |







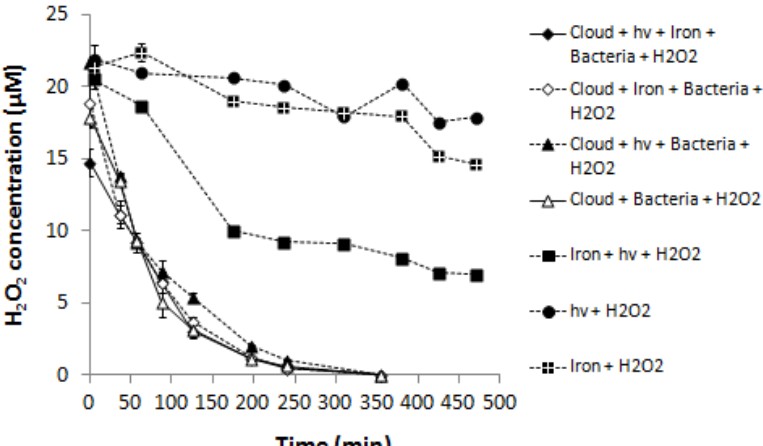


**Figure 1: Evolution of H$_2$O$_2$ concentration as a function of time (min) under abiotic conditions: Light + Fe-[EDDS] (black square), Light (black circle), Fe-[EDDS] (black square with white cross) and biotic conditions: Light + Fe-[EDDS] + *Pseudomonas graminis* (13b-3) (black diamond), Fe-[EDDS] + *Pseudomonas graminis* (13b-3) (white diamond), Light + *Pseudomonas graminis* (13b-3) (black triangle), *Pseudomonas graminis* (13b-3) (white triangle). Three replicates were done. Error bars (very low values) represent the standard errors (SEs) of the enzymatic assay (5%).**




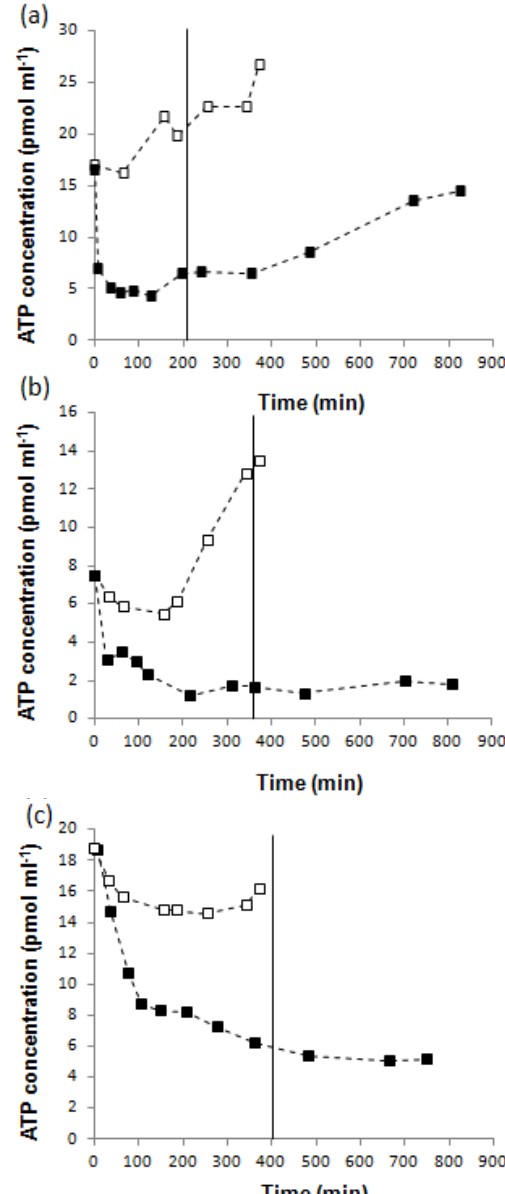



**Figure 2: ATP concentration (µM) as a function of time (min) in the presence (black square) or the absence (white square) of H$_2$O$_2$ for the three strains: (a) *Pseudomonas graminis* (13b-3), (b) *Pseudomonas syringae* (13b-2), (c) *Sphingomonas sp.* (14b-5).**

**The vertical bar illustrates the time corresponding to the total degradation of H$_2$O$_2$.**









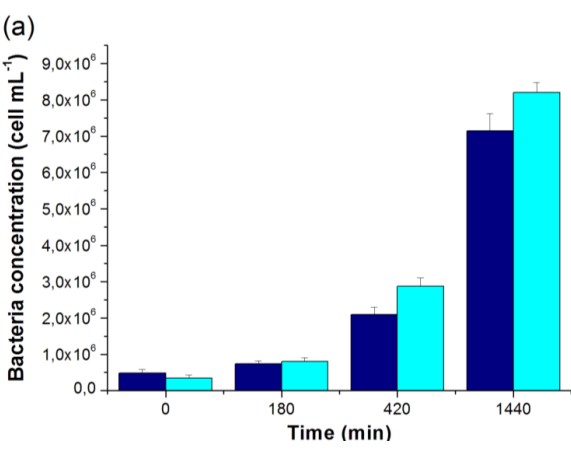

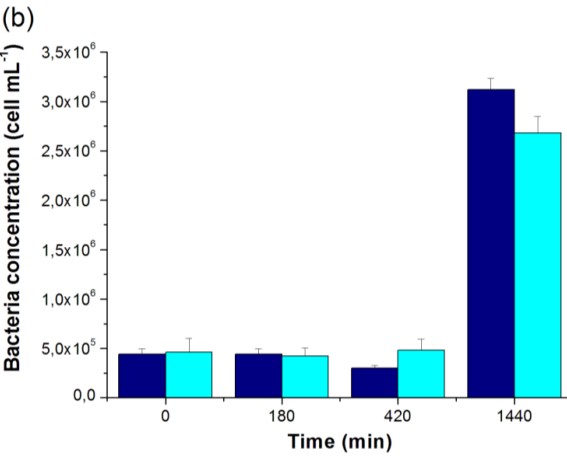

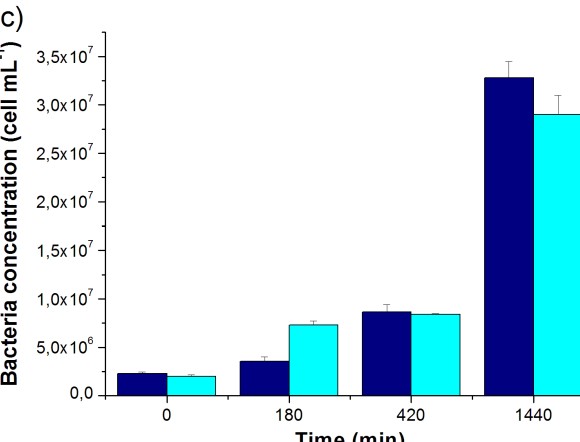

**Figure 3: Bacterial cell numbers measured by plate-counting in the absence (light blue) and the presence (dark blue) of H₂O₂ at 20 µM for the three strains: (a) *Pseudomonas graminis* (13b-3), (b) *Pseudomonas syringae* (13b-2) and (c) *Sphingomonas sp.* (14b5). Error bars represent standard deviation from the means (n=3).**




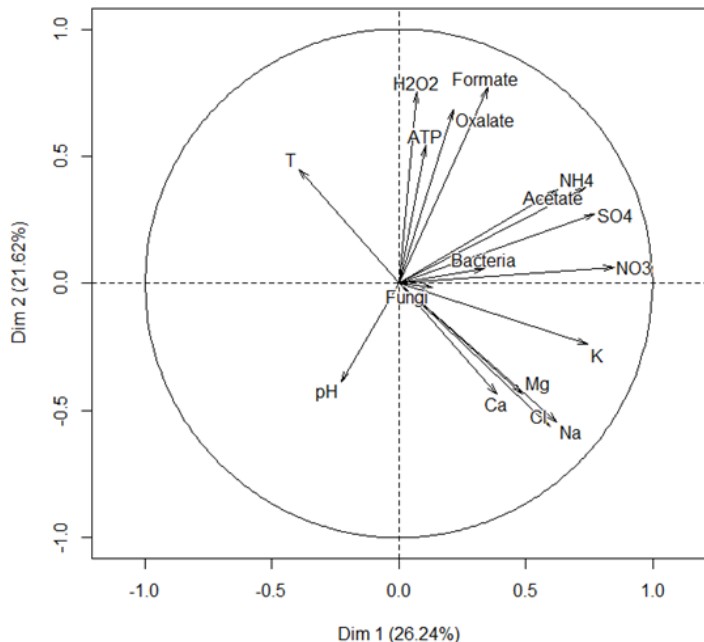


**Figure 4: Variables factor map (PCA) of the 37 cloud events on the plane PC1-PC2 based on 17 variables.**




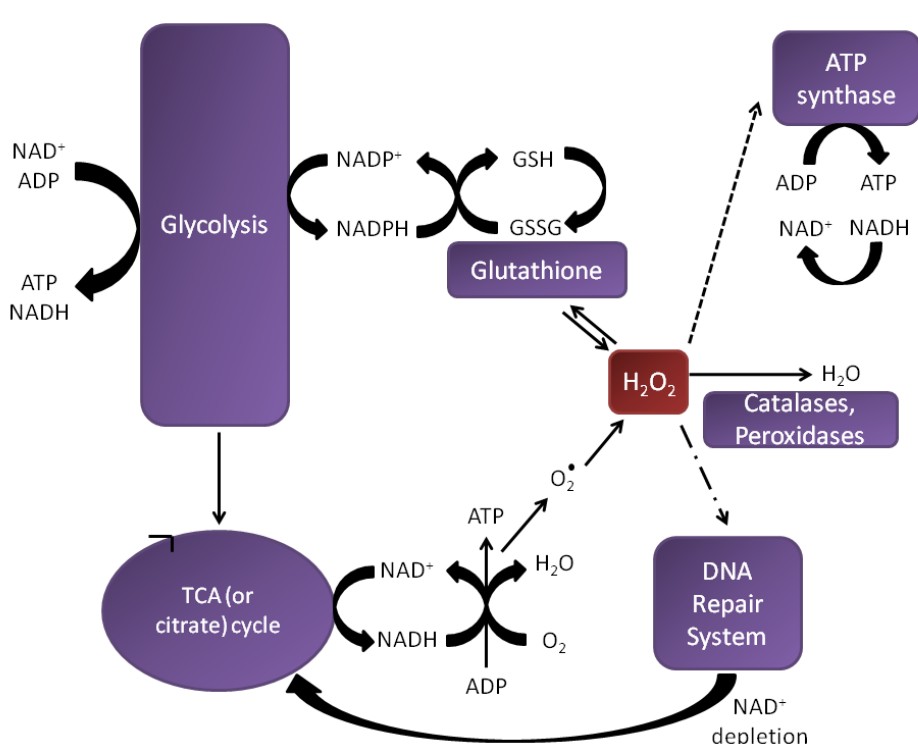


**Figure 5: Impact of H$_2$O$_2$ on cell metabolism and ATP concentration. Interconnection between ATP synthesis and**
**cellular redox potential (NAD$^+$/NADH, NADP$^+$/NADPH ratios). NAD$^+$ depletion related to DNA repair system.**
**Adapted from Oka et al. (2012).**

------▶    **Inhibition of ATP synthase**
—·—·—▶    **NAD$^+$  depletion related to DNA repair system**





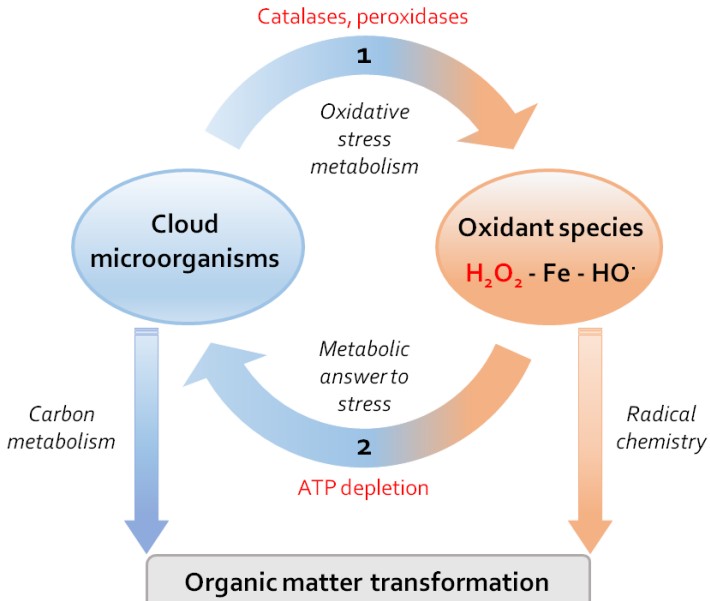


**Figure 6: Interaction between $H_2O_2$ and cloud microorganisms and its potential consequences on atmospheric chemistry. (1) Cloud microorganisms degrade $H_2O_2$ thanks to their catalases and peroxidases (oxidative stress metabolism) as a result it impacts the oxidant capacity of clouds. The concentration of radicals issued from $H_2O_2$ is decreased and radical chemistry is less efficient to transform the organic matter. (2) $H_2O_2$ impacts the energetic metabolism of microorganisms that react to this stress. The depletion of ATP modulates the global carbon metabolism of the microorganisms, and consequently the transformation of the organic matter. These processes are modulated by the $H_2O_2$ concentration that varies depending on atmospheric scenari.**

527