# Peer review of "H2O2 modulates the energetic metabolism of the cloud microbiome"

_Atmospheric Chemistry and Physics, 2017_

## Referee Comment (RC1) · Anonymous Referee #1 · 4 Sep 2017

Review Comment on "H2O2 modulates the energetic metabolism of the cloud microbiome"

General Comments This manuscript describes experiments and statistical analysis of field data that indicate that cloud bacteria have a strong impact on the loss of H2O2 from cloud water and that the bacteria exhibit depleted ATP after exposure to H2O2. The work is important because it provides additional evidence that the presence of living microorganisms in cloud water strongly affects the chemistry of the cloud water with implications for cloud processing and downstream outcomes. This work is novel and of high quality. I have provided specific and technical comments below.

Specific Comments Line 32: change to "formation and fate" to indicate formation and degradation may be affected

[Figure]

Line 76, 235: Here and elsewhere, I would suggest eliminating the use of "microflora" and use either "microbial community", "microorganisms", or "microbiome"

Line 114: 10481 g should be rounded to realistic significant figures

Line 116 – 118: Please add a citation for the technique

Line 120: use ". . . cloud water solution. . ."

Line 127: Please briefly state how the pH was adjusted.

Line 150-151: Please clarify that the H2O2 and iron complex were added at in situ cloud water concentrations, but all other constituents and bacteria were added at 10x the in situ concentration as stated in Lines 128-131. How does this concentration discrepancy affect the overall chemical reactivity of the cloud water medium as compared to in situ cloud water? How does this difference affect the activity of the microbes? Is there any concern that the microorganisms would be less stressed or vulnerable under the artificial conditions than actual cloud water conditions?

Line 164-166: This passage is not very clear with respect to language and technical aspects and needs to be re-written. What is "affline function"?

Line 166: Clarify how the initial degradation rate was calculated. Via the first two time points? Or other?

Line 174-175: What is the fixed part of the sampler? What alcohol was used? How does alcohol vapor affect cloud water chemistry as the samples are collected?

Line 180-206: This entire passage is redundant. This passage does not represent results. Please eliminate or work relevant parts into the Introduction, Methods or Discussion.

Line 220-221: Redundant.

Line 222-225: Is there any significance to the fact that the Sphingomonas isolate is less

active on H2O2 or that Sphingomonas and Pseudomonas 13b-2 seem not to recover with respect to the ATP concentration as well as Pseudomonas 13b-3? Could the authors discuss further?

Line 248: Which previous conditions are referred to here?

Line 263-275: This passage is either restating the Methods, or should be moved to the Methods. The Methods should include how data were collected and how statistical analyses were performed. Here it might be better to discuss the final set of data that resulted – i.e. Line 268 – 269 where it is explained how many events were selected for use. Then followed by the presentation of the PCA results.

Also, here and in the Methods it would be good to state how many sampling events were available. Then it could be stated that 37 events (of xx total) were selected after the constraints (e.g. no more than 10 percent of missing values) were applied.

Line 268: It is not entirely clear exactly what the 10 percent refers to. Does this mean that no more than 10 percent of data for any specific sample or any specific parameter was missing?

Line 310: Since the specific transcriptomic /metabolomic response of the microorganisms was not determined, the authors should indicate that the organisms "likely" or "probably" responded to the conditions using the mechanisms stated.

Line 324: avoid "very" and other qualitative wording

Line 327-332: This passage is not clear. Do you mean that formate metabolism could be inhibited by presence of H2O2? Please expand this discussion a little more to make the intended points.

Table 1: What is the rationale for the number of significant figures shown in each case. Should they be different for different data sets?

Line 333-334 and Figure 6 legend: Please edit to indicate that this is a hypothesized

mechanism. Since the actual response of cells was not measured, these mechanisms cannot be known with certainty.

Line 342: It would be good to examine the response of the organisms on a transcriptomic basis as well to confirm what genes are expressed in response to the H2O2 stress. Technical Corrections

Line 40: use "parameters"

Line 60: use "...A few decades ago, living microorganisms were observed in cloud water..."

Line 62: use "nutrient"

Line 64: change "Few" to "Several"

Line 69: "...to efficiently degrade..."

Line 70: eliminate "to" and "to the"

Line 70: eliminate "have"

Line 79: use "radiation"

Line 81: eliminate the first occurrence of "the"

Line 87: eliminate "the"

Line 88: instead of "Thanks to the fact that..." use "Because..."

Line 90: eliminate "the"

Line 91: eliminate the first occurrence of "of"

Line 104-106: This sentence should be re-written. Something like "It is crucial to consider all sinks and sources of H2O2, especially in atmospheric chemistry models, since H2O2 impacts many relevant processes in the atmosphere."

Line 114: "g" should be italicized

Line 121: eliminate the space after "concentrations"

Line 129-130: use ". . .the bacterial cell concentration..."

Line 134: replace "consisted" with "were performed"

Line 139: add a space between the number value and the unit

Line 142 and elsewhere: use "rpm"

Line 164-166: This passage is not very clear with respect to language and technical aspects and needs to be re-written. What is "affline function"?

Line 168: Eliminate "The". Add the company for R.

Line 170: use "less than" instead of "inferior"

Line 174: use "sterilized beforehand"; replace "during" with "for"

Line 232 and elsewhere: use "within the same order of magnitude"

Line 233: replace "than" with "of"

Line 234: use ". . .separately analyze. . ."

Line 76, 235: Here and elsewhere, I would suggest eliminating the use of "microflora" and use either "microbial community", "microorganisms", or "microbiome"

Line 235: use "clouds"

Line 236: eliminate both "the"s

Line 245: use "strain"

Line 254-257: Redundant and restates methods. Eliminate the first two sentences and replace the next two with something like—"Results for the number of culturable bacteria in the presence or absence of H2O2 are shown in Figure 3. "

Line 260: replace "was multiplied" with "increased"

Line 287: comma after "ATP"

Line 288: use "less than" instead of "inferior"

Line 292: replace "as" with "since"

Line 304: replace "to" with "at"

Line 312: use "reported"

Line 330: eliminate the second occurrence of "the"

Figure and Tables For figures and tables, I would suggest using the following wording:

"Values shown are averages of triplicates plus/minus one standard deviation"

"Symbols are averages of triplicates and error bars represent the standard error. Where error bars do not appear they are smaller than the symbol"

―――――――――――――――――――――――――

---

## Referee Comment (RC2) · Anonymous Referee #2 · 13 Sep 2017

The authors present data meant to demonstrate the impact of H2O2 on the metabolism of bacteria in cloud water. The dataset is probably valuable but I find that the data analysis and presentation of the manuscript require major revision before it will be suitable for publication in ACP.

The authors should comment on the important differences that exist between the laboratory setup and the cloud droplet environment, namely due to the much larger volume in the laboratory. How many bacteria can we expect to live in one cloud droplet? How is bacterial population growth in a cloud droplet different from in the laboratory studies discussed here (do we even know the nature of this difference?)?

In the studies described here, while bacteria metabolism impacts the concentrations of trace species (and vice versa), the number of bacteria in the sample is also growing

(i.e., Figure 3). The different solutions studied showed different growth profiles, as evidenced in Figure 3 - and these growth profiles are no doubt different from what would happen in the much smaller volume of a cloud droplet. Data regarding the kinetic processing of an atmospheric trace species by bacteria in a growing population is not useful, and even misleading, for atmospheric chemists who are the readership of this journal, unless the growth process can be decoupled from the chemical processing rates. One way to do this after the fact would be by normalizing the rate data by the number of bacteria in the sample at each time point. The data should be re-analyzed with this fundamental issue in mind.

The literature review in the Introduction section consists mostly of a discussion of this group's prior work. More of an effort should be made to place this study in the context of the broader scientific literature.

Finally, the language throughout the manuscript and the abstract needs editing. In many instances the language is too vague or informal for a scientific publication. The paper also needs to be edited carefully for English grammar (especially subject-verb disagreement in multiple places in the manuscript).

---

## Author Comment (AC1) · 6 Oct 2017

Review Comment on "H2O2 modulates the energetic metabolism of the cloud microbiome"

General Comments This manuscript describes experiments and statistical analysis of field data that indicate that cloud bacteria have a strong impact on the loss of H2O2 from cloud water and that the bacteria exhibit depleted ATP after exposure to H2O2. The work is important because it provides additional evidence that the presence of living microorganisms in cloud water strongly affects the chemistry of the cloud water with implications for cloud processing and downstream outcomes. This work is novel and of high quality. I have provided specific and technical comments below.

Answer: First of all we would like to thank Referee #1 for his great interest in our work and for all the remarks he made to improve the manuscript, including corrections of the English language. Changes are highlighted in yellow in the revised manuscript (see supplementary file)

Specific Comments:

Comment: Line 32: change to "formation and fate" to indicate formation and degradation may be affected C1 Answer: Ok, done

Comment: Line 76, 235: Here and elsewhere, I would suggest eliminating the use of "microflora" and use either "microbial community", "microorganisms", or "microbiome" Answer: Ok, done

Comment: Line 114: 10481 g should be rounded to realistic significant figures Answer: Replaced by "around 10000 g"

Comment: Line 116 – 118: Please add a citation for the technique Answer: Marie, D., Brussaard, C.P.D., Partensky, F., Vaulot, D. : Flow cytometric analysis of phytoplankton, bacteria and viruses, Robinson, J.P., Ed. Curr. Protoc. Cytom., John Wiley & Sons, 11.11, 1-15, 1999.

Comment: Line 120: use "... cloud water solution..." Answer: Ok, done

Comment: Line 127: Please briefly state how the pH was adjusted. Answer: This sentence was added lines 156-157 "Finally, the obtained solution was adjusted to pH 6 as necessary with a few drops of the solutions of NaOH or $H_2SO_4$ used for the preparation of the marine artificial cloud water solution".

Comment: Line 150-151: Please clarify that the $H_2O_2$ and iron complex were added at in situ cloud water concentrations, but all other constituents and bacteria were added at 10x the in situ concentration as stated in Lines 128-131. How does this concentration discrepancy affect the overall chemical reactivity of the cloud water medium as compared to in situ cloud water? How does this difference affect the activity of the microbes? Is there any concern that the microorganisms would be less stressed or vulnerable under the artificial conditions than actual cloud water conditions?

Answer: As stated in Lines 199-202 of the original manuscript: "Hydrogen peroxide and iron complex (Fe-[EDDS]) were added or not to the solution in the incubators. These two compounds are present in marine cloud water collected at the PUY station at average concentrations of 7.5 $\mu$M (with a dispersion of mean values ranging from 0.1 – 20.8 $\mu$M) for H2O2 and 0.5 $\mu$M (with a dispersion of mean values ranging from BDL. ïĂ■ 4.9) for Fe(III) (Deguillaume et al., 2014)". Therefore the concentrations used here for marine cloud water are thus compatible with real values at the PUY station when multiplied by a factor ten (20 $\mu$M for H2O2 and 4 $\mu$M for Fe(III) complex).

We have moved this paragraph to the Material and Method section lines 126-135. Of course any change in the concentrations can affect cloud metabolism, we show here that the major factor impacting ATP content is H2O2 while the presence of Fe(III)-EDDS does not modify this effect to a great extent. H2O2 concentration can indeed vary with atmospheric scenarios as stated in the introduction and discussion. This is what we have demonstrated from statistical analyses (Figure 4 and p values), ATP concentrations are correlated to H2O2 concentrations.

Comment: Line 164-166: This passage is not very clear with respect to language and technical aspects and needs to be re-written. What is "affline function"? Answer: It is actually an "affine "function (mathematical function)

Comment: Line 166: Clarify how the initial degradation rate was calculated. Via the first two time points? Or other?

Answer: The text was changed as follows lines 187-190: The processing of data was done with the Origin 6.1 software. The graphs representing the hydrogen peroxide concentration decrease as a function of time were plotted. The degradation rates have been calculated from the initial slopes (the first five time points i.e. between 0 and 2 hours) normalized with the concentrations of cells. During these two hours no cell

growth was observed.

Comment: Line 174-175: What is the fixed part of the sampler? What alcohol was used? How does alcohol vapor affect cloud water chemistry as the samples are collected?

Answer: Only the metal sheet is disinfected by alcohol (70%) and washed with sterile water consequently alcohol has no impact on cloud water chemistry. The collector itself is not treated with alcohol and is autoclaved and kept sterile until use. The text was modified as follows lines 194-196: The detachable part of the impactor was sterilized beforehand by autoclave at 121°C for 20 min and the fixed part was rinsed with alcohol at 70° and then with sterile water just before sampling.

Comment: Line 180-206: This entire passage is redundant. This passage does not represent results. Please eliminate or work relevant parts into the Introduction, Methods or Discussion.

Answer: We fully agree with the reviewer, so we have moved and merged this section with the Material and Method section lines 109-174 as follows:

Material and methods 2.1. Description of the microcosms Microcosms were designed to simulate as much as possible the water phase of cloud waters. They provide the opportunity to work under artificial solar light condition and also in the presence of microorganisms. For irradiation condition the bioreactor was equipped with lamps that emit UV-radiation (Sylvania Reptistar; 15 W; 6500 K; UVA (up to 30%), UVB (up to 5%)) to mimic solar light measured directly in clouds at the PUY station (Fig. SM1). The incubation flasks were Pyrex crystallizers covered with a Pyrex filter and equipped with Teflon tubes of 8 mm Ø plugged with sterile cotton, letting air and light pass while for dark conditions they were amber Erlenmeyer flasks. All incubation flasks contained 100 mL of artificial cloud solution under agitation (130 rpm). This solution was mimicking cloud chemical composition from cloud samples classified as "marine" following the work from Deguillaume et al. (2014) at the PUY station. The major part of the collected cloud samples were classified as marine (52%) supporting our choice for the artificial cloud composition. For biotic conditions, the flasks were inoculated at 106 bacterial cells per mL (Vaïtilingom et al., 2013). The three selected bacterial strains belonging to the Gamma-Proteobacteria (Pseudomonas) and Alpha- Proteobacteria classes (Sphingomonas) were isolated from cloud water and are representative of the genera most frequently found in cloud water samples (Vaïtilingom et al., 2012) collected at the PUY site. Depending on the conditions, hydrogen peroxide and iron complex (Fe-[EDDS]) were added or not to the solution in the incubators. These two compounds are present in marine cloud water collected at the PUY station at average concentrations of 7.5 $\mu$M (with a dispersion of mean values ranging from 0.1 – 20.8 $\mu$M) for H2O2 and 0.5 $\mu$M (with a dispersion of mean values ranging from BDL. – 4.9) for Fe(III) (Deguillaume et al., 2014). In the cloud aqueous phase, Fe(III) may be complexed by organic compounds. Recently, it has been hypothesized than iron can be chelated by other organic ligands of biological origin (Herckes et al., 2013; Herrmann et al., 2015), and in particular by siderophores (Vinatier et al., 2016) that are ligands characterized by high complexing constants (K>1020). Fe-[EDDS] was chosen as an iron(III) complex model because this ligand has a complexing constant for iron very close to the values for siderophores. Moreover, it is known to be stable at the working pH of 6.0 and because its chemistry has been studied in details by Li et al. (2010). In addition, the working temperature was fixed at 17°C which is the average temperature of cloud samples in summer.

2.2 Bacterial strains and growth conditions Pseudomonas graminis, 13b-3, DQ512786; Pseudomonas syringae, 13b-2, DQ512785, Sphingomonas sp., 14b-5, DQ512789 were grown in 10 mL of R2A medium (Reasoner and Geldreich, 1985) under stirring (200 r.p.m) at 17°C for approximately 17 h, 24 h or 48 h, depending on the strain. Cells in the exponential growth phase were collected by centrifugation for 3 min at around 10000 g. The supernatant was removed and the bacterial pellet was suspended and washed twice with an artificial cloud solution (2.2). The bacterial cell concentration was estimated by optical density at 575 nm to obtain a concentration close to 106 cell mL-1.

Finally, the concentration of cells was precisely determined by flow cytometry analysis (BD Facscalibur Becton-Dickinson; $\lambda$exc= 488 nm; $\lambda$em = 530 nm) using a method based on the addition of a fluorochrome (SYBR-green) for their counting (Marie et al., 1999).

2.3 Biodegradation assays Biodegradation assays were performed in marine artificial cloud water solution that mimics real cloud conditions as described in Vaïtilingom et al. (2011). Stock solutions were prepared with the following concentrations: 200 $\mu$M for acetic acid (CH3COOH; Acros organics), 145 $\mu$M for formic acid (HCOOH; Fluka), 30 $\mu$M for oxalic acid (H2C2O4;Fluka), 15 $\mu$M for succinic acid (H6C4O4; Fluka), 800 $\mu$M for ammonium nitrate (H4N2O3; Fluka), 100 $\mu$M for magnesium chloride hexahydrate (MgCl2, 6H2O; Sigma-Aldrich), 50 $\mu$M for potassium sulfate (K2SO4; Fluka), 400 $\mu$M for calcium chloride dihydrate (CaCl2, 2H2O; Sigma-Aldrich), 2000 $\mu$M for sodium chloride (NaCl; Sigma-Aldrich), 1100 $\mu$M for sodium hydroxide (NaOH; Merck), 315 $\mu$M for sulfuric acid (H2SO4; Sigma-Aldrich). Finally, the obtained solution was adjusted to pH 6 as necessary with a few drops of the solutions of NaOH or H2SO4 used for the preparation of the marine artificial cloud water solution and sterilized by filtration (Polyethersulfone membrane, 0.20 $\mu$m; Fisher Scientific) before use. The artificial cloud water solution was ten times more concentrated than a real cloud water solution in order to stabilize the pH. This was also the case for bacteria concentration because the bacteria/substrate ratio should be kept identical to that of real cloud. Indeed, it has been demonstrated that if this ratio is maintained, the degradation rate remains constant (Vaïtilingom et al., 2010). The equipment was sterilized by autoclaving at 121°C for 20 minutes and all manipulations were performed under sterile conditions. Biodegradation assays were performed in marine artificial cloud solutions inoculated with bacterial cells and incubated in a bioreactor (Infors HT Multitron II) at 17°C in the presence or absence of hydrogen peroxide solution, of iron complex solution and under irradiation or obscurity condition. At regular intervals, samples were taken and stored at -20 °C. Hydrogen peroxide solution was prepared from a commercial solution (H2O2, 30%; not stabilized Fluka Analytical). 1:1 stoichiometry iron complex solution was prepared from iron (III) chloride hexahydrate (FeCl3, 6H2O; Sigma-Aldrich) and from (S,S)- ethylenediamine-N,N'-disuccinic acid trisodium salt (EDDS, 35% in water). The hydrogen peroxide solution and the iron complex solution were freshly prepared before each experiment and the final working concentrations were fixed at 20 $\mu$M and 4 $\mu$M respectively, in agreement with the real concentrations detected in samples collected at the PUY station multiplied by a factor ten when median values measured in marine cloud waters are considered (Deguillaume et al., 2014).

Comment: Line 220-221: Redundant. Answer: We agree with the referee, it was changed line 231 by "For the biotic conditions, the initial biodegradation rates are summarized in Table 1(b)."

Comment: Line 222-225: Is there any signïficance to the fact that the Sphingomonas isolate is less active on H2O2 or that Sphingomonas and Pseudomonas 13b-2 seem not to recover with respect to the ATP concentration as well as Pseudomonas 13b-3? Could the authors discuss further?

Answer: Of course each individual strain can behave slightly differently, the tested strains here are model strains. In principle as Sphingomonas are well represented in the cloud microbiome this could impact the whole system. However we have shown that the H2O2 biodegradation rates measured here are within the same order of magnitude as those measured with real cloud water (Vaitilingom et al 2013), so it proves that this impact is not so high. In addition, concerning the ATP concentrations, our in-lab experiments are validated by the statistical analyses performed with the 37 cloud events (figure 4). Also the growth rate of Sphingomonas is not changed in the presence of H2O2 (Figure 3). In conclusion the differences between Pseudomonas and Sphingomonas have no major consequence on the global response of the system.

Comment: Line 248: Which previous conditions are referred to here?

Answer: We refer to the experiments in the presence of H2O2 alone. The sentence has been modified lines 257-259 as follows: Complementary experiments were performed with incubations of the cells in the presence or absence of light and/or iron complex (Fe-[EDDS]) under conditions similar to that described previously in the presence of H2O2 alone.

Comment: Line 263-275: This passage is either restating the Methods, or should be moved to the Methods. The Methods should include how data were collected and how statistical analyses were performed. Here it might be better to discuss the final set of data that resulted – i.e. Line 268 – 269 where it is explained how many events were selected for use. Then followed by the presentation of the PCA results. Also, here and in the Methods it would be good to state how many sampling events were available. Then it could be stated that 37 events (of xx total) were selected after the constraints (e.g. no more than 10 percent of missing values) were applied. Line 268: It is not entirely clear exactly what the 10 percent refers to. Does this mean that no more than 10 percent of data for any specific sample or any specific parameter was missing?

Answer: We took into account the referee's remark and moved this paragraph to the methods section lines 192-210 as follows:

"2.6 Cloud sampling and statistical analysis Cloud water sampling was performed on the summit of the PUY station (summit of the puy de Dôme, 1465 m a.s.l., France) which is part of the atmospheric survey networks EMEP, GAW, and ACTRIS. The detachable part of the impactor was sterilized beforehand by autoclave at 121°C for 20 min and the fixed part was rinsed with alcohol at 70° just before sampling. Between 2004 and 2013, 89 cloud events were collected at the PUY station. The origin of these clouds can be analyzed according to their back trajectories in four sectors (North/West, South/West, West and North/East). They can be also considered in four different categories considering their chemical composition (marine, continental, highly marine and polluted) as described in Deguillaume et al. (2014). Various parameters were measured including ATP, bacteria and fungi concentration, inorganic and organic species concentration (H2O2, SO42-, NO3-, Cl-, acetate, formate, oxalate, Na+, NH4+, Mg2+, K+, Ca2+), temperature and pH (see Table SM1 for details). More information about

[Figure]

the cloud sample collection is given in Deguillaume et al. (2014). These data were used in this study to achieve statistical analyses. R software 3.1.2 was used to carry out principal component analysis (PCA). The data of 37 cloud events (of 89 total) were selected after the constraints related to this statistical analysis (e.g. the cloud events with more than 10 percent of missing values (parameters) were not considered) were applied. In addition, statistical significance test was evaluated using PAST software (Hammer et al., 2001). Mean difference was considered to be statistically significant for a p-value less than 0.05."

Comment: Line 310: Since the specific transcriptomic /metabolomic response of the microorganisms was not determined, the authors should indicate that the organisms "likely" or "probably" responded to the conditions using the mechanisms stated.

Answer: We agree with the referee, this is only a hypothetical mechanism. The text has been changed as follows lines 312-315:

This reveals that microorganisms are able to manage the stress induced by $H_2O_2$ through their metabolism. It is likely that they could respond using enzymes involved in $H_2O_2$ degradation (e.g. catalases, peroxidases, etc.) and other typical antioxidant molecules (glutathione, etc.).

Comment: Line 324: avoid "very" and other qualitative wording Answer: OK changed to "high"

Comment: Line 327-332: This passage is not clear. Do you mean that formate metabolism could be inhibited by presence of $H_2O_2$? Please expand this discussion a little more to make the intended points.

Answer: We agree it was not clear enough, so we have added this sentence Lines 336-338:

"Indeed formate contributes to the anti-oxidant strategy of this bacterium to supply NADH which is known to be decreased under oxidative conditions, formate helps thus to control the cellular redox potential (see Fig. 5)."

Comment:Table 1: What is the rationale for the number of significant figures shown in each case. Should they be different for different data sets? Answer: Sorry but I do not understand this question.

Comment: Line 333-334 and Figure 6 legend: Please edit to indicate that this is a hypothesized mechanism. Since the actual response of cells was not measured, these mechanisms cannot be known with certainty.

Answer: We agree with the reviewer. The text and the Fig. 5 legend have been changed as follows:

Lines 318-322. "Fig. 5 illustrates how H2O2 could affect the concentration of ATP in the cells. First H2O2 could directly inhibit the ATP synthase, a membrane protein synthetizing ATP from ADP (Tamarit et al 1998). Second H2O2 could impact different metabolic pathways which are interconnected including glutathione metabolism, glycolysis, TCA cycle and DNA repair system."

Legend: Figure 5: Hypothetical mechanism that could explain the impact of H2O2 on cell metabolism and ATP concentration. Interconnection between ATP synthesis and cellular redox potential (NAD+/NADH, NADP+/NADPH ratios).

Comment: Line 342: It would be good to examine the response of the organisms on a transcriptomic basis as well to confirm what genes are expressed in response to the H2O2 stress.

Answer: This is a good suggestion; we have changed the text line 346 as follows: To go further in the understanding of the modulation of the metabolic pathways (including carbon, nitrogen, amino-acids or sugars) induced by H2O2, a combined metabolomic and transcriptomic approach could be used.

Comment :Technical Corrections: Answer: We thank the reviewer for these valuable corrections. Changes have been made in the revised manuscript.

Line 40: use "parameters" Line 60: use "...A few decades ago, living microorganisms were observed in cloud water..." Line 62: use "nutrient" Line 64: change "Few" to "Several" Line 69: "...to efiňĄciently degrade..." Line 70: eliminate "to" and "to the" Line 70: eliminate "have" Line 79: use "radiation" Line 81: eliminate the first occurrence of "the" Line 87: eliminate "the" Line 88: instead of "Thanks to the fact that..." use "Because..." Line 90: eliminate "the" Line 91: eliminate the first occurrence of "of" Line 104-106: This sentence should be re-written. Something like "It is crucial to considerall sinks and sources of H2O2, especially in atmospheric chemistry models, since H2O2 impacts many relevant processes in the atmosphere." Line 114: "g" should be italicized Line 121: eliminate the space after "concentrations" Line 129-130: use "...the bacterial cell concentration..." Line 134: replace "consisted" with "were performed" Line 139: add a space between the number value and the unit Line 142 and elsewhere: use "rpm" Line 164-166: This passage is not very clear with respect to language and technical aspects and needs to be re-written. What is "afine function"? Answer: "affine function" Line 187-190: The processing of data was done with the Origin 6.1 software. The graphs representing the hydrogen peroxide concentration decrease as a function of time were plotted. The degradation rates have been calculated from the initial slopes (the first five time points i.e. between 0 and 2 hours) normalized with the concentrations of cells

Line 168: Eliminate "The". Add the company for R. Line 170: use "less than" instead of "inferior" Line 174: use "sterilized beforehand"; replace "during" with "for" Line 232 and elsewhere: use "within the same order of magnitude" Line 233: replace "than" with "of" Line 234: use "...separately analyze..." Line 76, 235: Here and elsewhere, I would suggest eliminating the use of "microflora" and use either "microbial community", "microorganisms", or "microbiome" Line 235: use "clouds" Line 236: eliminate both "the"s Line 245: use "strain" Line 254-257: Redundant and restates methods. Eliminate the first two sentences and replace the next two with something like "Results for the number of culturable bacteria in the presence or absence of H2O2 are shown in Figure 3. " Line 260: replace "was multiplied" with "increased" Line 287:

[Figure]

comma after "ATP" Line 288: use "less than" instead of "inferior" Line 292: replace "as" with "since" Line 304: replace "to" with "at" Line 312: use "reported" Line 330: eliminate the second occurrence of "the" Figure and Tables For figures and tables, I would suggest using the following wording: "Values shown are averages of triplicates plus/minus one standard deviation" "Symbols are averages of triplicates and error bars represent the standard error. Where error bars do not appear they are smaller than the symbol"

Please also note the supplement to this comment:
https://www.atmos-chem-phys-discuss.net/acp-2017-581/acp-2017-581-AC1-supplement.pdf

[Figure]

**Supplement:**

[revised manuscript text omitted]

**The vertical bar illustrates the time corresponding to the total degradation of $H_2O_2$.**

[Figure]

[Figure]

[Figure]

**Figure 3: Bacterial cell numbers measured by plate-counting in the absence (light blue) and the presence (dark blue)**

**of $H_2O_2$ at 20 μM for the three strains: (a)** *Pseudomonas graminis* **(13b-3), (b)** *Pseudomonas syringae* **(13b-2) and (c)**

*Sphingomonas sp.* **(14b5). Error bars represent standard deviation from the means (n=3).**

[Figure]

**Figure 4: Variables factor map (PCA) of the 37 cloud events on the plane PC1-PC2 based on 17 variables.**

[Figure]

**Figure 5: Hypothetical mechanism that could explain the impact of $H_2O_2$ on cell metabolism and ATP concentration.**
**Interconnection between ATP synthesis and cellular redox potential ($NAD^+$/NADH, $NADP^+$/NADPH ratios). $NAD^+$**
**depletion related to DNA repair system. Adapted from Oka et al. (2012).**
------▶ **Inhibition of ATP synthase**
—·—·—·▶ **$NAD^+$ depletion related to DNA repair system**

[Figure]

**Figure 6: Interaction between $H_2O_2$ and cloud microorganisms and its potential consequences on atmospheric chemistry. (1) Cloud microorganisms degrade $H_2O_2$ thanks to their catalases and peroxidases (oxidative stress metabolism) as a result it impacts the oxidant capacity of clouds. The concentration of radicals issued from $H_2O_2$ is decreased and radical chemistry is less efficient to transform the organic matter. (2) $H_2O_2$ impacts the energetic metabolism of microorganisms that react to this stress. The depletion of ATP modulates the global carbon metabolism of the microorganisms, and consequently the transformation of the organic matter. These processes are modulated by the $H_2O_2$ concentration that varies depending on atmospheric scenari.**

---

## Author Comment (AC2) · 6 Oct 2017

Comment: The authors present data meant to demonstrate the impact of H2O2 on the metabolism of bacteria in cloud water. The dataset is probably valuable but I find that the data analysis and presentation of the manuscript require major revision before it will be suitable for publication in ACP.

Answer: First of all we would like to thank Referee #2 for all his comments that should help to improve the manuscript.Changes are highlighted in yellow in the revised

manuscript (see supplementary file).

Comment: The authors should comment on the important differences that exist between the laboratory setup and the cloud droplet environment, namely due to the much larger volume in the laboratory. How many bacteria can we expect to live in one cloud droplet? How is bacterial population growth in a cloud droplet different from in the laboratory studies discussed here (do we even know the nature of this difference?)?

Answer: Actually nobody really knows the absolute difference between in-lab and droplet conditions for the growth of bacteria. We suspect that one droplet contains one bacterium as bacteria can be considered as CCN and thus form a droplet. If we consider doubling times measured with a few strains isolated from cloud waters (Amato , PhD thesis, 2004 ) they varied from 5h to 20h at 17°C (average temperature in summer time at the PUY station) and from 16 h to 45 hours at 5°C (average winter temperature). Also during incubation at 17°C of a real cloud sample containing the whole microbiome and chemical composition of cloud water we measured an increase of cell concentration from 105 bacteria /mL to 106 bacteria /mL within 100 hours (Amato et al., Atmos. Chem. Phys, 2007, 5253-5276).

These experiments suggest that, depending on the strains and the temperature, and considering the duration of a cloud for about 2 days, the bacteria could divide from one to ten times.

However we would like to point out that this debate, although it represents still an open question, is out of the scope of this paper. The objective of the experiments presented in Figure 3 was only to demonstrate that bacteria did not die although their ATP content was drastically decreased. Growth measurement is a global proxy to attest the viability of the cells.

Comment: In the studies described here, while bacteria metabolism impacts the concentrations of trace species (and vice versa), the number of bacteria in the sample is also growing (i.e., Figure 3). The different solutions studied showed different growth

profiles, as evidenced in Figure 3 - and these growth profiles are no doubt different from what would happen in the much smaller volume of a cloud droplet. Data regarding the kinetic processing of an atmospheric trace species by bacteria in a growing population is not useful, and even misleading, for atmospheric chemists who are the readership of this journal, unless the growth process can be decoupled from the chemical processing rates. One way to do this after the fact would be by normalizing the rate data by the number of bacteria in the sample at each time point. The data should be re-analyzed with this fundamental issue in mind.

Answer: We fully understand the remark of the reviewer; this indicates that we did not clearly define the objective of determining rates of degradation of H2O2. I think that the interpretation of Table 1 might be misleading. We have to clarify different points:

First the biodegradation rates have been calculated from the initial slopes (the first five time points i.e. between 0 and 2 hours) normalized with the concentrations of cells. Looking at Figure 3 it is clear that none of the bacteria are dividing (growing) during that 2 hour period (< 200 min.). Consequently the comparison of the abiotic and biotic degradation rates during that period is not altered by a change in the number of cells.

The purpose of the experiments performed in a microcosm with different conditions (bacteria or not, iron, light. . .) was not to measure degradation rates that will be directly implemented in atmospheric models or to quantify the relative contribution of abiotic versus biotic routes in atmospheric chemistry. In the past we have done it and indeed we have expressed the rates of biodegradation in mol. h-1.cell-1 (Vaitilingom et al. Appl. Environ. Microb., 2010, 76, 23-29 ; Vaitilingom et al. Atmos. Chem. Phys., 2011, 11, 8721-8733 ; Husarova et al. Atmos. Environ., 2011, 45, 6093-6102). If atmospheric chemists want to integrate growth in their model, they have to increment the number of cells at each time step of the calculation in the model. But this is out of the scope of this paper.

The major goal of this paper was to show that H2O2 modulates the ATP concentration

of the cloud microbiome. Experiments in laboratory help understanding what the major factor influencing ATP depletion was. The development of the microcosms allowed us to separate the different factors (Fe, H2O2, light, . . .) and to conclude that only H2O2 concentration was important. To raise such a conclusion it was necessary to first validate that the microcosms used could mimic as much as possible cloud conditions. The idea to measure degradation rates in these microcosms was to get values (or rather "orders of magnitude") to be compared with those obtained with more realistic conditions. Our results show that the degradation rates measured are within the same order of magnitude that those obtained with real cloud water samples (Vaitilingom et al, Proc. Natl. Acad. Sci USA, 2013, 110, 559 564) and validate thus these microcosms.

The link between H202 and ATP concentrations observed under laboratory conditions was also validated in real cloud events using statistical analyses.

We hope that these explanations will help reviewer 2 to better understand our purpose. To make the objective of the work clearer and avoid any misleading in interpretation, we have changed the text as follows:

This sentence was added in the Material and Method section 188-190: "The biodegradation rates have been calculated from the initial slopes (the first five time points i.e. between 0 and 2 hours) normalized with the concentrations of cells. During these two hours no cell growth was observed. "

This sentence was deleted line 238: "These results show that artificial light and Fe-[EDDS] and thus HO● radicals have no effect on H2O2 biodegradation".

We have modified this section line 238-246: The selected strains all degrade H2O2 within the same order of magnitude (average value for the three strains and for the condition with iron and light 1.76 10- 9 mol L-1 s-1 and with iron without light 1.40 10-9 mol L-1 s-1). In Vaïtilingom et al. (2013), the biodegradation rates of H2O2 were found within the same order of magnitude (average value for two distinct clouds with light 0.98 10-9 mol.L-1 s-1 and without light 0.29 10-9 mol L-1 s-1). The results obtained

are within the same order of magnitude of values in real cloud environment thereby validating our microcosm conditions. This demonstrates that under our experimental conditions, the selected strains degrade H2O2 like the microbiome of real clouds. In addition it validates our approach to separately analyse the influence of each parameter (Fe, H2O2, light,. . .) on the microbial energetic state metabolism detailed in the next section.

Comment: The literature review in the Introduction section consists mostly of a discussion of this group's prior work. More of an effort should be made to place this study in the context of the broader scientific literature.

Answer: 49 references are cited, from them 16 are from our group.

Among these 16 papers one is a review (Delort et al 2017) citing thus a lot of other references and 9 of them refer to the impact of cloud microorganisms on atmospheric chemistry. Actually, except the group of Ariya (which is cited) no other group works on this specific topic related to the interaction between microorganisms and cloud chemistry.

To extend this aspect to the air, we have added a reference of Krumins, V.; Mainelis G., Kerkhof, L.J.; and Fennell, D.E. Substrate-dependent rRNA production in an airborne bacterium. Environmental Science and Technology Letters, 2014, 9, 376-381.

The other citations of our group concern mainly measurements at the PUY station which are necessary for this work.

Most of the other references are centered on cloud chemistry and have been chosen to focus on hydrogen peroxide as it is the main purpose of this paper. Some of them are reviews (Gunz and Hoffmann 1990, Vione et al 2003) also citing many other papers.

To make the atmospheric chemistry context even wider we have added:

the extensive review of Herrmann H, Schaefer T, Tilgner A, Styler SA, Weller C, Teich M, et al. Tropospheric aqueous-phase chemistry: kinetics, mechanisms, and its

coupling to a changing gas phase. Chem Rev. 2015; 115:4259–334.

And theses references:

Li, J., Wang, X., Chen, J., Zhu, C., Li, W., Li, C., Liu, L., Xu, C., Wen, L., Xue, L., Wang, W., Ding, A. and Herrmann, H.: Chemical composition and droplet size distribution of cloud at the summit of Mount Tai, China, Atmospheric Chem. Phys. Discuss., 1–21, doi:10.5194/acp-2016-1175, 2017.

Shen, X., Lee, T., Guo, J., Wang, X., Li, P., Xu, P., Wang, Y., Ren, Y., Wang, W., Wang, T., Li, Y., Carn, S. A., and Collett, J. L.: Aqueous phase sulfate production in clouds in eastern China, Atmospheric Environment, 62, 502-511, https://doi.org/10.1016/j.atmosenv.2012.07.079, 2012.

Arakaki, T.; Anastasio, C.; Kuroki, Y.; Nakajima, H.; Okada, K.; Kotani, Y.; Handa, D.; Azechi, S.; Kimura, T.; Tsuhako, A. A general scavenging rate constant for reaction of hydroxyl radical with organic carbon in atmospheric waters. Environmental Science & Technology ,2013 , 47 (15), 8196-8203.

Hems, R.F.; Hsieh, J.S.; Slodki, M.A.; Shouming, Z. ; Abbatt, J.P.D. Suppression of OH Generation from the Photo-Fenton Reaction in the Presence of $\alpha$-Pinene Secondary Organic Aerosol Material. Environmental Science and Technology Letters Article ASAP DOI: 10.1021/acs.estlett.7b00381

Wei, M., Xu, C., Chen, J., Zhu, C., Li, J., and Lv, G.: Characteristics of bacterial community in cloud water at Mt Tai: similarity and disparity under polluted and non-polluted cloud episodes, Atmos. Chem. Phys., 17, 5253-5270, https://doi.org/10.5194/acp-17-5253-2017, 2017.

Comment: Finally, the language throughout the manuscript and the abstract needs editing. In many instances the language is too vague or informal for a scientific publication. The paper also needs to be edited carefully for English grammar (especially subject-verb disagreement in multiple places in the manuscript).

Answer: We agree with reviewer 2 that the language should be improved. Hopefully reviewer 1 carefully corrected the manuscript and helped us to improve its quality.

Please also note the supplement to this comment:
https://www.atmos-chem-phys-discuss.net/acp-2017-581/acp-2017-581-AC2-supplement.pdf

———————————————————

---

## Author Response (AR2)

**Co-Editor Decision: Publish subject to minor revisions (review by editor)** (23 Oct 2017) by Alex Huffman

Comments to the Author:

Authors,

After reading through the referee comments and your responses, I am confident that the manuscript will soon be acceptable for final publication. There are a few areas that I would like you to improve upon somewhat more before final acceptance, and I've listed these below. Hopefully these comments will be relatively efficient for you to process edits for. There are also still some areas that require minor English language edits, but these will likely be corrected during the copy-editing process by the Copernicus staff.

Alex Huffman

General comments:

Comment 1: Section 2.1, Description of the microcosms – I think the addition of this section was important, but I'm still a bit confused by how these microcosms were produced in your experiments. I suggest adding a sentence after the first in the section to say something like: "Microcosms were developed by …" and then follow with an few-sentence, explicit overview of how you got droplets of cloud-like water. It wasn't clear to me whether the water was collected from clouds and processed in some way or synthetically produced. Make this very clear at the beginning of this section before you move into the details of the radiation that was supplied to the microcosms.

Answer: We understand that this section was not clear enough and confusing. We have completely changed this section as follows. We moved the description of the cells and their growth conditions in section 2.1 (initially section 2.2) and merged "the description of the microcosm" (initially section 2.1) and "the biodegradation assays" (initially section 2.3) in the same section (now section 2.2 "Incubations in microcosms"). We also added new sentences.

**2.1 Bacterial strains and growth conditions**

[revised manuscript text omitted]

Comment 2: For Figures 1,2, and 4, I think it would benefit the reader to consider using color as a part of the traces/markers. Since color figures can be reproduced in the final version at no additional cost, I think this edit would improve readability, especially in the relatively complicated Figure 1. You might consider coloring in such a way as to make one theme of colors to be biotic and another theme to be abiotic, etc.

Answer: thank you for this remark, all the figures are in color now.

Connected to this comment, e.g. at L222 where Figure 1 is discussed, the interpretation of the figure takes some time for the reader. I suggest specifically referring to which trace you discuss by the color that you change it to in the final form, i.e. "the degradation of hydrogen peroxide is clearly effective … (Fig. 1, blue trace)"

Answer: we took into account this remark (lines 222 and 231).

Comment 3: Reviewer #2 suggested major manuscript revision, including several specific areas of improvement. My feeling is that some of these suggestions for improvement are reasonable, but can be handled with only mild additional discussion. One area relates to their first major comment about the differences between the laboratory setup and the 'real' cloud water environment. I agree with your response that a full analysis of this is beyond the scope of the manuscript. However, I suggest taking some of your response to the referee's question and including these major ideas somewhere in the manuscript – probably the final discussion, Section 4.

Comment 4: The same response is true for the second comment from Referee #2 ("In the studies described here …"). I think it would be worthwhile to add an additional sentence or two of discussion regarding how an atmospheric chemist might treat or use these data. In this context it is fine to say what would need to be done before it could be modeled or scaled, and how it might be important or complicated. I encourage you to use thoughts you have already formulated and put into the response document. Take the most important of these and add a few ideas to the discussion.

Answer to comments 3 and 4: In order to take into account some of Referee #2 comments ,we added this paragraph to the discussion section (lines 327-337):

"…induced by $H_2O_2$, then ATP is depleted, and finally all the metabolic pathways involving these compounds are impacted and a complete change in the metabolome can be expected.

The measurements preformed in microcosms do not reproduce what is really occurring in cloud droplets. First incubations were performed with artificial cloud water and model strains, nevertheless the obtained results were consistent with those obtained with real cloud water samples. Second the potential growth of microorganisms during a cloud event could also modify transformation rates, this is only realistic for long cloud lifetimes ( > 24 hours). Finally experiments were performed under bulk conditions and not with individual cloud droplets, only models can take into account the complexity of cloud conditions, in particular the multiphase aspect of cloud chemistry. To go further and integrate biodegradation rates in atmospheric chemistry models, complementary experiments should be performed and biodegradation rates should be expressed as $mol^{-1}.cell^{-1}.h^{-1}$.

However the most important result of this work was to show the correlation between $H_2O_2$ concentrations and ATP concentrations. This result obtained under our microcosm conditions was confirmed using data measured in real cloud samples that experienced multiphase and real cloud conditions. Indeed, we have shown, thanks to statistical analyses, that there was also a high correlation between $H_2O_2$ and ATP concentrations in real cloud…"

Technical comments and typos:
L61: Typo – remove "have been"
Done
L106: New sentence – I would suggest moving this sentence a bit one sentence higher into the paragraph so that the sentence beginning "This work will …" can be the last sentence of the introduction.

Done
L114: I was a bit confused by how to interpret the "up to 30%" comment about the UV radiation. 30% of what, of the light energy emitted from the chosen source? Please clarify this.
Answer: This information is not important (%); we have changed the text to " Sylvania Reptistar; 15 W; 6500 K " (line 114 )

Figure 1 caption: I suggest changing to: "Where error bars are not visible they are smaller than the symbol." Also, the two sentence previous to this are almost exactly redundant. Keep only one of those two sentences.
Answer:  We agree this is redundant. We have changed the legend by your proposition "Where error bars are not visible they are smaller than the symbol"

Figures 1, 2, and 4 are produced at low resolution. After acceptance, please make sure to submit higher resolution versions of these figures.

Answer: The image format of these figures has been changed (Export), the resolution is quite correct now.

Table 1: Is there a reason why no standard errors are reported for section (a), abiotic degradation? It would be better if these were included.

Answer : it was a mistake, we have added the standard  errors.